# Linking Housing Conditions and Energy Poverty: From a Perspective of Household Energy Self-Restriction

**DOI:** 10.3390/ijerph19148254

**Published:** 2022-07-06

**Authors:** Keyu Chen, Chao Feng

**Affiliations:** School of Economics and Business Administration, Chongqing University, Chongqing 400030, China; kkchenkeyu@163.com

**Keywords:** housing conditions, energy poverty, energy restriction, logit models

## Abstract

Affordable and warm housing is a basic household living need, which is closely related to human health and well-being. This study attempts to establish the link between household housing conditions and energy poverty in China from the perspective of energy self-restriction using logit and mediation models based on microdata. The results report that: (1) households are more likely to be exposed to energy poverty if they live in larger, older, poorly insulated houses, without basic energy service equipment, and in rental housing; (2) the area of residence and energy installations are the main characteristics that distinguish energy poverty from non-energy poverty; (3) the link between housing conditions and energy poverty is reinforced by the psychology and behavior of households, with those living in poor conditions tending to restrain their energy consumption, thus worsening their energy poverty situation.

## 1. Introduction

Energy is a basic human need that is closely linked to human health and well-being. The United Nations Sustainable Development Goals (SDGs) incorporate energy targets, proposing the goal of ensuring access to affordable, reliable, sustainable and modern energy for all [1]. However, for a variety of reasons, many households are facing energy poverty through under-use and over-burdened energy. As reported, about 759 million people worldwide lack access to electricity in 2019. By the end of 2020, more than 25 million people in developing countries in Asia and Africa could lose the ability to afford a basic package of electricity services [2]. Additionally, the European Commission reports that more than 34 million people in the EU experience energy poverty to various degrees [3]. Addressing energy poverty is a global public policy challenge due to the complexity, diversity and invisibility of its manifestations, and the damage it can do to human well-being and physical and mental health.

The earliest research on energy poverty dates back to the 1980s, when Bradshaw and Hutton (1983) described it as the inability to obtain adequate warmth in the home and proposed three policies to alleviate energy poverty by increasing income to cover energy costs [4]. Subsequently, Boardman (1991) described energy-poor households from the perspective of energy expenditure as ‘disproportionate expenditure’, with households identified as energy-poor if they spent more than 10% of their income on access to energy services [5]. Since then, more research has been carried out on the definition [6,7,8], influencing factors [9,10] and consequences [11,12] of energy poverty.

As housing and energy are closely linked, the housing factor becomes an important perspective in examining energy poverty. Decent, warm, comfortable homes require basic energy services, including space heating and cooling, lighting, water heating, cooking and electricity [13]. On the one hand, the necessary energy needs of a household are highly correlated with the housing characteristics, such as the energy efficiency [14,15,16], building envelope [17,18] and thermal insulation [19,20]. In general, inefficient energy systems [21] and insulation can increase the energy costs of a household to some extent. When households cannot afford high energy costs, they choose to reduce their energy consumption. When household energy consumption falls below the amount of energy required to maintain basic decent living conditions, energy poverty occurs. On the other hand, when housing costs account for a high share of household income, this affects the ability of households to pay their energy bills and households are forced to reduce their energy expenditure, resulting in a situation of the under-consumption of energy [22]. Furthermore, conversely, unfavorable housing conditions and poor housing quality are often associated with energy poverty, for example, energy-poor households may often experience damp houses and mold on the walls and floors [23].

In China, as urbanization continues, more and more residents are beginning to cluster in cities and some low-income households are facing housing affordability and energy affordability challenges. In urban areas, some older neighborhoods have poor energy infrastructure (e.g., electricity, gas and heating) and also suffer from inefficient buildings, increasing the risk of household energy poverty. In rural areas, a large number of houses are detached and poorly insulated, making it difficult for some households to maintain comfortable thermal conditions in their homes. At the same time, due to low income and poor basic energy facilities, some households have difficulty accessing modern energy services and rely heavily on inefficient traditional energy sources. Since housing and energy issues are related to the standard of living, health and well-being of occupants, both housing and energy poverty are pressing livelihood issues to be addressed. However, few studies have analyzed the link between housing and energy poverty in China [24,25,26]. We therefore hope to identify the characteristics of energy-poor households in China from a housing perspective and seek new pathways to alleviate energy poverty.

Figure 1 depicts the direct and indirect links between household housing conditions and energy poverty. We argue that housing conditions can construct direct links to energy poverty through their own physical attributes, such as energy efficiency, thermal insulation and energy appliances. In addition, we hypothesize that occupants’ behavior (choices) can strengthen the link between household housing conditions and energy poverty. The occupants’ choices in specific housing conditions may play a role in the energy poverty of households. That is, housing conditions can construct an indirect link with energy poverty through occupants’ behavior.

Therefore, we attempt to link housing conditions to energy poverty in order to identify the main housing characteristics of energy-poor groups and to analyze the role of housing factors in their energy poverty to inform the development of targeted policies. In this study, we first use multiple indicators to identify energy poverty and classify energy-poor households into three categories. Secondly, we use a logit model to construct the link between housing conditions and energy poverty and obtain the housing characteristics and energy poverty characteristics of energy-poor households in each category. Finally, we discuss the role of household psychology and behavior in mediating the link between housing conditions and energy poverty in terms of energy consumption constraints using a mediation model. The results of this study enrich the findings on the housing characteristics of energy-poor households in China. Furthermore, by constructing direct and indirect links between housing conditions and energy poverty, our results provide new insights into how energy poverty can be considered from a housing perspective.

The rest of the study is structured as follows: Section 2 contains the key literature review relating to housing conditions and energy poverty; Section 3 describes the data and methodology of the study; Section 4 presents and discusses the empirical results; and Section 5 provides the conclusions.

## 2. Literature Review

Initial research on energy poverty focused on the definition and measurement of energy poverty [5,6,7,8] and this has been extensively researched and debated [27,28,29]. Subsequent studies have broadened the scope of energy poverty research and started to focus on the drivers [9,10] and the consequences [11,12,30,31,32] of energy poverty. Due to the strong link between housing and energy, a number of studies have discussed the relationship between housing stock characteristics and energy poverty.

The energy efficiency of housing is considered as an important influence on energy poverty. Hills [6] identifies fuel prices, low income and energy efficiency as the three main drivers of energy poverty. The energy performance of a building affects a household’s energy demand and is a contributor to energy poverty [33]. In general, households whose dwellings are less energy efficient are at greater risk of exposure to energy poverty and are likely to face more severe energy poverty [14,15,16,34,35]. A study carried out in the UK suggests that wealthier middle-income households may fall into energy poverty as a result of living in relatively inefficient homes [36]. Housing energy efficiency is highly correlated with indoor thermal conditions, with low-energy performance dwellings generally having lower indoor temperatures [20,37]. A study of energy poverty in low-income households showed that residential energy efficiency can cause large differences in heating costs and increase the energy burden of inefficient households [38].

Housing insulation is an important component of residential energy efficiency. There is a significant relationship between housing insulation and energy poverty [17,18]. The installation of double glazing and insulated roofs indicate a more insulated home. Studies have shown that improving housing insulation is effective in improving the indoor environment of homes, reducing condensation, mold and dampness problems [39], as well as improving the thermal comfort of homes [19,20].

Another condition associated with residential energy efficiency is the age of the dwelling. Older homes are often associated with conditions such as poor insulation and outdated energy systems [10], which together lead to energy poverty in households. Older dwellings tend to have higher heat loss [20] and poorer thermal regulation [9]. Households may be at higher energy risk if they live in older homes [9,18,23,33,40,41]. Furthermore, in terms of energy costs, households may pay higher energy bills for normal energy use because of their older homes [36], resulting in high energy costs as a proportion of income.

In addition to the energy efficiency attributes of housing, the other physical attributes (size and type) of housing may also be correlated with energy poverty. The link between house size and energy poverty may be based on energy bills [42,43], with larger homes being colder than smaller ones [20] and households being burdened with heavier energy costs in order to keep rooms warm, increasing the risk of household energy vulnerability exposure [10,36]. However, some studies have also reported the opposite result, with households living in small spaces being more vulnerable to energy instability [21,44], while households with larger homes are less vulnerable to energy poverty [45]. In addition, some studies have discussed the characteristics of house types of energy-poor households, and it is found that households living in detached, semi-detached dwellings are more likely to be exposed to energy poverty [40,46,47], which may be associated with higher levels of heat loss [48]. Additionally, the area of residence may also influence energy poverty [49,50].

Energy-poor households can also be affected by the energy system and energy installations in their homes. The installation of a heating system in a home can be effective in improving the indoor environment, enhancing the comfort of the home and alleviating energy poverty [18,39]. Efficient heating systems can reduce household energy costs and reduce the risk of energy poverty in households [18,21,51]. In addition, the household’s energy equipment and type of energy source are also important factors influencing energy poverty [9,36].

In terms of the economic attributes of housing, housing tenure is regarded as a driver of energy poverty [10,50]. It has been shown that energy-poor households are predominantly renters [9,44], and living in rented housing increases the risk of energy poverty compared to owner-occupied dwellings [21,40,51,52]. One possible explanation is that rented housing suffers from low energy inefficiency. There are insufficient incentives for landlords to retrofit their homes for energy efficiency, while tenants have limited rights and insufficient incentives [53] to implement effective energy efficiency improvements and retrofitting measures in their homes. Rental housing therefore has more serious energy efficiency problems [54]. On the other hand, renting is a manifestation of lower household income [55] and, in order to limit expenditure, choosing low-cost housing is a necessity, which may result in more energy consumption due to the poor thermal performance of the housing [56]. Low income and high energy consumption therefore make renting households more vulnerable to energy poverty. Furthermore, from an income distribution perspective, when housing costs account for a large proportion of household income, households are constrained in the amount of money they can spend on energy consumption and therefore fall into energy instability [45].

Current research on energy poverty in China falls into three main categories of topics: (1) assessing energy poverty in China [57,58]; (2) exploring the factors [59,60,61] influencing energy poverty from macro and micro perspectives; and (3) analyzing the impact of energy poverty on income [62], health [63,64], personal development [65], child well-being [66] and psychological status [67,68] from a household perspective. However, only a small number of studies have discussed the link between buildings and energy poverty [24,25,59]. Therefore, we hope to dissect the energy poverty of Chinese households from a housing perspective and provide some new ideas for effective energy poverty alleviation.

## 3. Materials and Methods

### 3.1. Energy Poverty Indicators

The energy-poor group identified is closely related to the set of energy poverty indicators. There can be differences between the groups of energy-poor households identified by different energy poverty indicators, and they often cover different households. That is, the households identified as energy-poor, according to different energy poverty indicators, are often in different energy situations and have different energy challenges. In this study, we measure the energy poverty of households from the perspective of income and expenditure. The target group identified is those energy vulnerable households that carry a heavy energy burden and do not consume enough energy. For them, paying for their entire energy needs is difficult for various reasons (e.g., low income, high energy prices), and therefore their energy needs are often restricted.

In order to fully consider the risk of energy poverty for households of different sizes, the concepts of equivalent income and equivalent energy costs are introduced in the construction of the energy poverty indicator. When it comes to the actual use of energy, as family members live in the same space, some energy services are shared, such as lighting, space heating and cooling. Therefore, when the household size increases, the energy costs of the household do not increase correspondingly but have a scale effect on energy use. Following the new equivalence of scale criteria released by OECD (2011) [69], household income and energy costs are divided by the square root of household size to obtain equivalent income and equivalent energy costs, respectively.

In this study, three energy poverty indicators are employed to measure household energy poverty. The first energy poverty indicator (EPI1) is constructed based on the equivalent income, equivalent energy costs household and equivalent income poverty threshold, identifying households that cannot afford basic energy consumption due to low income. The equivalent income poverty threshold is 60% of the median equivalent income of all households interviewed [70]. Following the after-fuel-cost poverty approach (AFCP) [6], we identify households as income poor if their income before energy costs is already below the poverty threshold or households whose income after energy costs is below the poverty threshold (income before energy costs is above the poverty threshold), as shown in Equation (1). The former are income-poor households, and the latter are energy-induced-income-poor households (or energy vulnerable), both of which are classified as energy-poor.
(1)$$\begin{array}{ccc} \mathrm{Energy}\;\mathrm{poverty} & = & \mathrm{Equivalent}\;\mathrm{income}\;\mathrm{before}\;\mathrm{or}\;\mathrm{after}\;\mathrm{energy}\;\mathrm{costs}\\ & < & 60\%\;\mathrm{of}\;\mathrm{the}\;\mathrm{median}\;\mathrm{equivalent}\;\mathrm{income} \end{array}$$

The second energy poverty indicator is based on the household’s energy expenditure and reflects the actual energy consumption of the households in need. A household is considered to be in energy poverty if it consumes too little energy to meet its needs for energy services to maintain basic living requirements. Following the indicator for measuring energy poverty proposed by Energy Poverty Observatory (EPOV)—low absolute energy expenditure (M/2), a household is classified as energy-poor if its energy expenditure is less than half that of the median household [8], as shown in Equation (2).
(2)$$\begin{array}{ccc} \mathrm{Energy}\;\mathrm{poverty} & = & \mathrm{Energy}\;\mathrm{expenditure}\\ & < & \frac{\mathrm{Median}\;\mathrm{household}\;\mathrm{energy}\;\mathrm{expenditure}}{2} \end{array}$$

The third energy poverty indicator (EPI3) is constructed on the basis of the first two energy poverty indicators. This indicator classifies the respondent households into four categories, namely (1) non-energy-poor; (2) IHCL: EPI1 identifies energy-poor and EPI2 identifies non-energy-poor; (3) ILCH: EPI1 identifies non-energy-poor and EPI2 identifies energy-poor; and (4) ILCL: dual identification as energy-poor. We assign values 0, 1, 2 and 3 to each of the above four categories in order. Figure 2 shows the distribution and proportion of households in each category.

### 3.2. Data

In this study, we employ cross-sectional data from the Chinese General Social Survey 2015 (CGSS2015) [71] to examine housing conditions and energy poverty in China. The Chinese General Social Survey (CGSS) is a nationwide survey project that systematically and comprehensively collects data at multiple levels of society, communities, households and individuals. In the CGSS2015 version, a new energy module (Section E) has been added to provide housing and energy information of selected households, such as economic and physical attributes of dwellings, energy consumption, energy devices and behavioral patterns of residential energy use. In the energy module, there are a total of 3557 sample households; however, the indicators for some households are not applicable to our research. After screening the applicable sample, we finally obtained 2402 observations that could be used for this study.

In order to comprehensively construct the link between housing conditions and energy poverty, we analyzed the housing conditions of households in terms of multiple dimensions, namely, area of residence, basic dwelling attributes, housing insulation, sunlight and basic energy equipment. The area of residence indicates whether the household lives in a rural or urban area. In general, households’ energy use may be limited by regional energy infrastructure and energy habits and there are differences in energy infrastructure and energy habits between rural and urban areas. Basic dwelling attributes include three dimensions: (1) housing tenure, which distinguishes whether the household is a homeowner or a renter; (2) housing size, to identify whether the household lives in an overcrowded space; and (3) dwelling age, to distinguish between dwellings built before 1980 and those built after 1980. In 1980, China started the process of housing reform, and the quality of housing was improved. Housing insulation is reflected by whether the home is fitted with double or triple glazing. Sunlight is measured by the number of hours of sunlight a household receives in summer and winter, respectively. Basic energy equipment describes the household’s use of energy devices related to basic energy services, namely (1) space heating and cooling, whether the household has heating or air conditioning equipment, and (2) water heating, whether the household uses a water heater to heat water.

In addition, we consider the moderating role of residents’ psychology and behavior in housing choice and energy consumption. In a further analysis, we introduce psychological feelings and modern energy service preferences of households to represent households’ subjective responses between housing conditions and energy poverty. The psychological perception of the household is represented by the level of happiness of the householder. Modern energy service preference is reflected by the sum of the number of basic domestic appliances in the household, including refrigerators, freezers, washing machines and dryers. A clearer description of the variables is presented in Table 1.

### 3.3. Methodology

Considering the energy poverty indicators used in this study, we use the binary logit regression model and multinomial logit regression model to examine the association between housing conditions and energy poverty. The advantage of the logit model is that by setting a baseline category, it is possible to directly compare characteristics of an outcome category with that of the baseline category, while the regression coefficients can provide information on the relative change in the probability of the outcome category to the baseline category when the explanatory variable (i.e., the characteristics under examination) change. In our study, we set non-energy-poor households as the baseline category and compare each category of energy-poor households with non-energy-poor households to effectively determine the link between housing conditions and energy poverty.

The binary logit regression model is applied to energy poverty identified using EPI1 and EPI2. We define the category of households as, *y_i_* = 0 if non-energy-poor and *y_i_* = 1 if energy-poor. The predicted probability of a household experiencing energy poverty is shown in Equation (3):(3)$$P\left(y_{i}=1|x\right)=\frac{\exp\left(x_{i}^{\prime }\beta_{1}\right)}{1+\exp\left(x_{i}^{\prime }\beta_{1}\right)}$$
where *x_i_* denotes a vector of housing conditions variables for household *i*, and $\beta_{1}$ denotes a vector of estimated coefficients (the logarithm of the odds ratio of a household experiencing energy poverty relative to non-energy poverty). In the subsequent analysis, the vector of estimated coefficients is converted to relative risk ratios (RRRs) using an exponential transformation.

Multinomial logit regression model is employed to analyze the relationship between housing conditions and energy poverty categories subdivided according to EPI3. The category *y_i_ = j* for households is defined as *y_i_* = 0 if EPI3 = 0; *y_i_* = 1 if EPI3 = 1; *y_i_* = 2 if EPI3 = 2 and *y_i_* = 3 if EPI3 = 3. The predicted probability of a household suffering from energy poverty is shown in Equation (4):(4)$$P\left(y_{i}=j|x\right)=\frac{\exp\left(x_{i}^{\prime }\beta_{j}\right)}{1+\sum_{j=1}^{3}\exp\left(x_{i}^{\prime }\beta_{j}\right)}\;\;\;\;\;\;\;\;\;\;j=0,\;1,\;2,\;3$$
where *y_i_* = 0 is the baseline category, *x_i_* denotes a vector of housing conditions variables for household *i* and $\beta_{j}$ refers to a vector of estimated coefficients for category *j* relative to the baseline category.

Further, we construct a mediation model of housing conditions and energy poverty. As we set the mediating variable as a continuous variable and the outcome variable as a categorical variable, the OLS and Logit techniques are used for the regression estimation, respectively. Given that the estimates come from different statistical machines, the traditional z-test approach is less appropriate. In this study, we follow the z-mediation test proposed by Iacobucci (2012) [72], who allows the combination of OLS and Logit regression results to test for mediating effects. The transmission path from the housing conditions variables to the mediating variables and from the mediating variables to the energy poverty categories are shown in Equations (5) and (6), respectively:
(5)$$m_{t}=\alpha_{0}+\alpha_{1}x_{i}+\varepsilon_{i}\;\;\;\;\;\;\;\;\;\;t=\;1,\;2$$
(6)$$P\left(y_{i}=j|x^{\prime }\right)=\frac{\exp\left(x_{i}^{\prime \prime }\beta_{j}\right)}{1+\sum_{j=1}^{3}\exp\left(x_{i}^{\prime \prime }\beta_{j}\right)}\;\;\;\;\;\;\;\;\;\;j=0,\;1,\;2,\;3$$
where *m_t_* denotes the vector of mediating variables and *x’_i_* refers to a vector of the combination of housing conditions and household behaviors variables. *ε_i_* is an error term.

## 4. Results and Discussion

### 4.1. Link between Housing Conditions and Energy Poverty

As mentioned in the previous section, a set of variables related to housing conditions is introduced into the logit regression models to examine the link between housing conditions and energy poverty. Table 2 shows the results of logit regression models based on the three energy poverty indicators.

The results in column (1) show the impact of housing conditions on the probability of household energy poverty, according to EPI1. Clearly, the area of residence is an important factor in causing energy poverty. The odds ratio of energy poverty to non-energy poverty is 0.3 times higher for those living in urban areas than for those living in rural areas. That is, households living in rural areas are more likely to experience energy poverty, which is similar to the existing findings [49]. In addition, the age of the dwelling significantly affects the probability of household energy poverty. As the RRR of housing age is significantly less than 1, this suggests that households living in older buildings are more likely to suffer from energy poverty. Considering the housing insulation, we find that households with double or triple glazing have a reduced probability of becoming energy-poor, as seen in other researches [9,18]. In addition, the longer the hours of sunlight in summer, the greater the probability of energy poverty. This outcome may be associated with other housing characteristics, such as single glazing. Specifically, only about 20.23% of the households with long summer sunlight hours in our sample are equipped with double or triple glazing. In terms of basic energy installations, the probability of a household falling into the energy poverty category increases if it does not have space heating or cooling equipment and if it cannot rely on a water heater to heat water. Therefore, based on EPI1, we can conclude that households living in rural areas, in old, poorly insulated houses with long hours of summer light, and without modern space heating and cooling and water heating equipment in the home, are more likely to be in the energy-poor group.

Column (2) displays the relationship between housing conditions and household energy poverty identified by EPI2. Similar to the estimates for EPI1, the probability of a household being identified as energy-poor increases if they live in a rural area and if the home does not have basic energy installations. Uniquely, the probability that a household is energy-poor decreases slightly if they enjoy a larger housing size. On the one hand, housing size is generally positively correlated with households’ financial level, and households living in larger houses may be better able to afford higher levels of energy consumption. Evidence from research in South Asia also supports this inference [45]. On the other hand, when households have more space per person, they may have to consume more energy to keep their rooms warm during the cold season, thus increasing their energy use. In addition, if households enjoy longer hours of sunlight in winter, they are less likely to be identified as energy-poor.

As EPI3 divides household energy poverty into three categories, we hope to be able to accurately identify the housing and energy poverty characteristics of households in each category and to focus on the most vulnerable groups. Prior to the analysis, we try to pre-determine the energy poverty characteristics of each category of households in order to better extract information from the regression results. Households in IHCL are better off financially, as their income remains above the poverty threshold after deducting energy costs but their energy consumption is below the basic level. Households in ILCH are not financially well off, but they do not suffer from insufficient energy consumption, and they may therefore bear a higher energy burden. In contrast, households in ILCL face both income poverty and energy shortage. Similar to existing research [10], we also identified a range of housing characteristics for different categories of energy-poor households.

Based on the results in column (3), an interesting finding is that households with double or triple glazing are more likely to fall into IHCL compared to non-energy-poor households. We can infer that households in IHCL may have better insulated houses and that the higher energy performance saves them some energy. However, in terms of other housing characteristics, households are more likely to fall into IHCL if they live in rural areas, live in more cramped housing, rent and lack modern energy devices for space heating, cooling and heating water. That is, these poor housing conditions can cause households to curtail their energy expenditure. The combination of both the energy performance of housing and the active limitation of households in terms of energy demand makes the household energy consumption below the basic level.

The results in column (4) show that a range of housing conditions increase the probability of a household falling into ILCH, including: living in rural areas, living in older dwellings, poorly insulated homes, with short hours of sunlight in winter, long hours of sunlight in summer, without space heating or cooling equipment and inability to rely on a water heater to heat water. In line with our preconceptions, the dwellings of such households are less energy efficient, and they therefore need to consume more energy to meet their energy needs to reach basic living conditions. Although their energy consumption is above the poverty level, households are likely to be under tighter financial pressure as their income is below the poverty line. At the same time, with limited disposable income, energy consumption may crowd out households’ consumption of other goods, thus creating poverty problems for households in other areas.

As households in ILCL face both income poverty and energy poverty, they are likely to be the most energy-vulnerable households and face chronic energy deficits. Based on the results in column (5), we find that households in ILCL have significantly smaller RRRs for the area of residence, age of dwellings and basic energy equipment variables compared to the other two categories. This means that households in ILCL are more likely to exhibit housing characteristics, such as living in rural areas, living in an older home and lacking basic household energy equipment. In other words, households in this category are more likely to live in poor housing conditions, and these factors significantly increase the risk of household exposure to energy poverty.

### 4.2. Marginal Effect

We use the Wald test to estimate whether housing conditions have a significant effect on energy poverty across categories, i.e., to test whether the coefficient on a characteristic variable is simultaneously non-zero across categories. The results of the Wald test indicate that area of residence, age of dwelling, housing insulation and basic household energy installations have a significant effect on energy poverty across categories. A new multinomial logit model is therefore constructed using these variables, and based on this we further analyze the marginal effects of housing conditions on the probability of a household’s energy poverty category.

Table 3 displays the marginal effects of housing conditions on the households in the energy poverty category. In terms of area of residence, households are significantly more likely to be non-energy-poor if they live in urban areas than in rural areas, by approximately 34.4%. Correspondingly, the probability of household energy poverty decreases, with households in the ILCH and ILCL categories being the most affected. Considering the age of dwelling, households living in older housing significantly increase the probability of households belonging to the ILCL category by about 3.2%. In addition, housing insulation is an important factor influencing the energy poverty category of households. According to the value of the marginal effect, the probability of a household falling into the IHCL category increases significantly by about 3.8% when the housing has higher insulation performance, while the probability of falling into the ILCH category decreases significantly by about 5.1%. We therefore tentatively conclude that households in the IHCL category are better insulated, while those in the ILCH category are worse insulated. Similar to the marginal effect of area of residence, households are significantly more likely to be non-energy-poor if they are equipped with basic energy devices, while they are significantly less likely to be energy-poor in all other categories. Based on the above analysis, we can conclude that area of residence and energy devices are the main characteristics that distinguish energy poverty from non-energy poverty.

Further, Figure 3 depicts the marginal effects of all housing conditions variables on the four outcome categories, helping us to better visualize the link between housing conditions and energy poverty. Obviously and visually, the fold depicting the marginal effect of non-energy-poor households consistently lies above the zero line, indicating that the probability of a household being non-energy-poor increases significantly as the quality of its housing improves. Therefore, we infer that good housing conditions are positively associated with non-energy poverty. Furthermore, we find that the line graphs depicting the marginal effects of the other three energy poverty categories lie largely below the zero line, with the exception of the IHCL category for the housing age and housing insulation performance variables. This suggests that housing characteristics effectively distinguish between energy-poor and non-energy-poor households. From this perspective, energy policies in relation to housing retrofit may help households to escape energy poverty to some extent. In addition, it is found that the line graph for the ILCH category deviates furthest from the zero line, meaning that households in this category are most affected by housing conditions.

### 4.3. Mediation Effect

Taking it even further, we rely on a mediation model to explore more links between housing conditions and energy poverty. Among other things, we focus on the controlling role of psychological and behavioral factors of households in energy use. Under specific housing conditions, households’ psychology and behavior may be influenced and respond in a biased manner to such objective housing characteristics. Hence, we want to investigate how households’ energy-related responses in the face of different housing conditions and whether this strengthens the link between housing conditions and energy poverty.

Table 4 presents the results of the mediation effects of households’ psychological perception and modern energy service preference. We first discuss the role of mediating variables in each energy poverty category. Based on the results of the tests, it is discovered that in IHCL, the link between housing conditions and energy poverty cannot be constructed through the household’s happiness, while the household’s energy preferences strengthen the link between housing conditions and energy poverty. However, in both ILCH and ILCL, both household happiness and modern energy preferences mediate the link between housing conditions and energy poverty.

Focusing on the mediating role of the psychological perception of households, it is noticed that housing insulation and basic energy installations are significantly and positively associated with the well-being of the home. That is, households in ILCH and ILCL experience a considerable increase in comfort and well-being when they are more insulated and have basic energy service installations. In particular, in ILCH, households feel slightly less happy if they live in an urban area. This can be explained by the fact that, on the one hand, households in ILCH are already at a poorer financial level, and if they live in urban areas, they may face higher energy and other consumption costs, resulting in a greater financial burden and therefore making them less happy. On the other hand, as households are below the poverty line level, the quality of their housing in urban areas may also be at a lower level, also making them less happy. In addition, the results for Path B suggest that when households’ well-being increases, the probability of them falling into ILCH and ILCL decreases and the effect is somewhat greater for ILCL. Therefore, we reason that households in ILCH and ILCL will be at a lower level of well-being compared to non-energy-poor households, and this low well-being may have an impact on their energy consumption decisions, especially for households in ILCL. Based on the above analysis, the role of household psychological perception mediators can be summarized as follows: for ILCH, living in urban areas, poorly insulated houses and lack of basic energy installations will increase the household’s discomfort, and, in order to alleviate this discomfort, the household may choose to consume more energy to maintain an adapted standard of living, and therefore the household faces a more severe energy and financial burden. For ILCL, household unhappiness and instability increase with poorly insulated homes and lack of basic energy installations, and this instability may cause households to limit energy expenditure, resulting in severe energy under-consumption.

Focusing on the mediating role of household energy preferences, the mediating test values indicate that households’ residential characteristics, including the area of residence, renting, age of dwelling, housing insulation, summer sunlight hours and basic energy installations can be more extensively linked to energy poverty through energy preferences. Based on the results of Path A, it is discovered that urban dwellers have higher modern energy preferences. We explain this phenomenon from two perspectives. On the one hand, urban dwellers have easier access to modern energy services due to better equipped energy-related infrastructure in cities and are therefore more likely to use modern energy. On the other hand, households living in rural areas may prefer to use traditional energy sources due to the ease of access, low costs and long-standing energy habits. Considering the impact of renting, households have a lower energy preference if they live in rented accommodation. The households that rent are generally low-income and are more prone to renting in poor housing conditions. Due to financial pressures, they may tend to limit their household energy use, but poor housing quality may require them to use more energy to meet basic living conditions, so they fall into a more difficult energy poverty situation. Furthermore, if households live in older homes, their modern energy preferences are significantly lower. Home insulation is obviously and positively associated with household energy preferences. In addition, we find a consistency of preference in the use of modern energy devices, with households having a higher preference for other modern energy services if they have basic energy devices installed.

The results for Path B show that households’ preferences for modern energy services have a notable negative relationship with energy poverty. As a household’s energy preferences increase, the probability of a household being energy-poor decreases. Among the three energy poverty categories, the probability changes the most in ILCL, i.e., households in ILCL are potentially associated with a lower level of energy preference, which is consistent with the energy poverty characteristics of households in ILCL. Therefore, based on the above analysis, we summarize the mediating role of energy preference as energy preference is positively related to the housing quality of households, living in poor housing environment reduces the energy preference of occupants, thereby discouraging energy consumption, and the households’ energy consumption cannot meet basic energy needs, resulting in an energy poverty problem. While this effect is strongest in ILCL, it is weakest in ILCH.

## 5. Conclusions

In this study, we attempt to establish a link between housing conditions and energy poverty by discussing the relationship between housing characteristics and energy poverty categories. Specifically, we first measure household energy poverty multidimensionally from the perspective of income and energy expenditure, using three indicators, namely AFCP, M/2 and the combination of AFCP and M/2. Households are classified into four categories: non-energy poverty, IHCL, ILCH and ILCL. We then use Logit models to explore the link between housing conditions and energy poverty and capture the housing characteristics and energy poverty characteristics of energy-poor households in each category. Finally, we discuss the role of household psychology and behavior in mediating between housing conditions and energy poverty from the perspective of energy consumption constraints using a mediation model, which strengthens the link between housing conditions and energy poverty.

The empirical results show that housing conditions including area of residence, housing size, dwelling age, housing tenure, housing insulation, hours of sunlight and basic household energy services are significantly correlated with energy poverty. Households are more likely to be exposed to energy poverty if they live in larger, older, poorly insulated houses that are not equipped with basic energy services and if they rent. Marginal effect analysis reports that area of residence and energy installations are the main characteristics that distinguish energy poverty from non-energy poverty. Compared to urban dwellers, those living in rural areas are at greater risk of energy poverty. At the same time, the basic energy equipment status of households is highly correlated with energy poverty status, with energy-poor households being partially deprived of basic modern energy services.

Further mediating results suggest that psychological and behavioral factors of households reinforce the link between housing conditions and energy poverty. It is noteworthy that there are differences in the responses of households within the different energy poverty categories. The responses of the most vulnerable households show that household unhappiness and instability increase with poorly insulated houses and a lack of basic energy installations, and that this instability may cause households to limit energy expenditure, resulting in significant under-consumption of energy. In addition, energy preferences are positively correlated with household housing quality, and living in a poor housing environment reduces occupants’ energy preferences, thereby discouraging energy consumption and causing the under-consumption of energy poverty.

Overall, housing conditions are consistent with energy poverty in that if households’ housing conditions are poor, the greater the households’ exposure to energy poverty, and this association is reinforced by the households’ subjective response—to limit energy consumption.

The results of this study provide new insights into how energy poverty can be considered from a housing perspective. It is worth noting that energy-poor households often live in poor housing conditions and relying solely on household decisions may make households’ energy poverty worse, making external intervention schemes and subsidy policies necessary. Programs to improve the energy efficiency of housing may be able to help some households escape energy poverty, but they are premised on the fact that we need to precisely identify the group that really needs this help, i.e., households that need to consume more energy to meet their basic living conditions because of the energy inefficiency of their housing. Furthermore, by linking the energy poverty characteristics of households to their housing conditions, we observe that one of the underlying drivers of housing conditions and energy poverty is low income. Therefore, helping the most energy-vulnerable households to increase their income or providing them with energy subsidies may be an effective way to alleviate their energy poverty situation.

It is important to note that the manifestation of energy poverty among households is highly heterogeneous, so targeted policy measures are key to effectively helping households to address and alleviate energy poverty. In formulating policies, policy makers need to consider the actual difficulties of households from a multidimensional perspective. Our results provide suggestions for identifying different categories of energy-poor households from a housing perspective, thus helping policy makers to formulate more targeted and effective policies. However, to some extent, our results are based on generalized data, which may allow us to ignore the details of smaller groups, such as extremely poor households, elderly groups and single-person households. Using more homogeneous data for analysis may be a further direction for our research.

## Figures and Tables

**Figure 1 ijerph-19-08254-f001:**
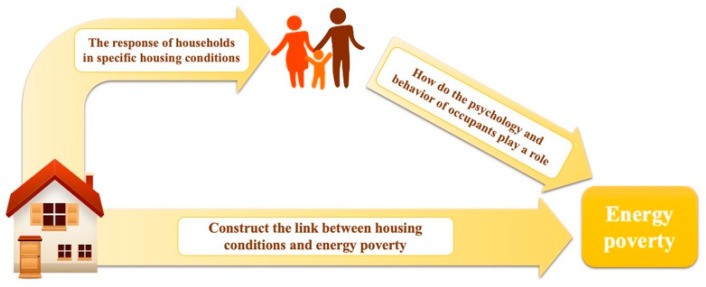
Direct and indirect links between housing conditions and energy poverty.

**Figure 2 ijerph-19-08254-f002:**
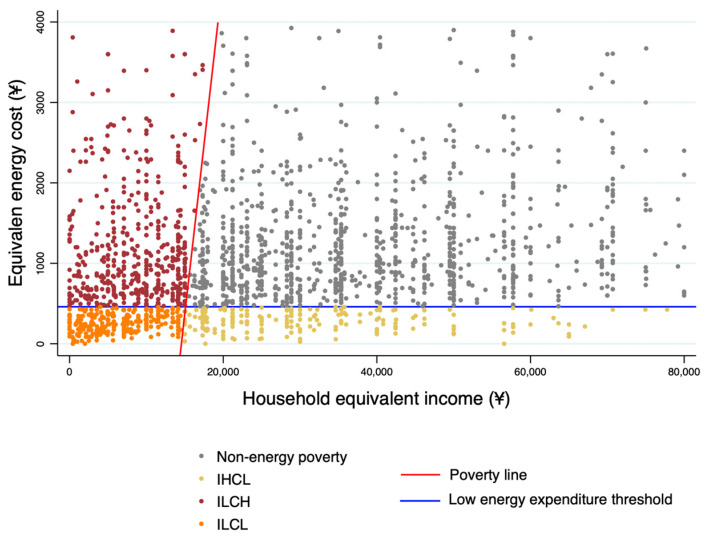
Distribution of households in different poverty categories.

**Figure 3 ijerph-19-08254-f003:**
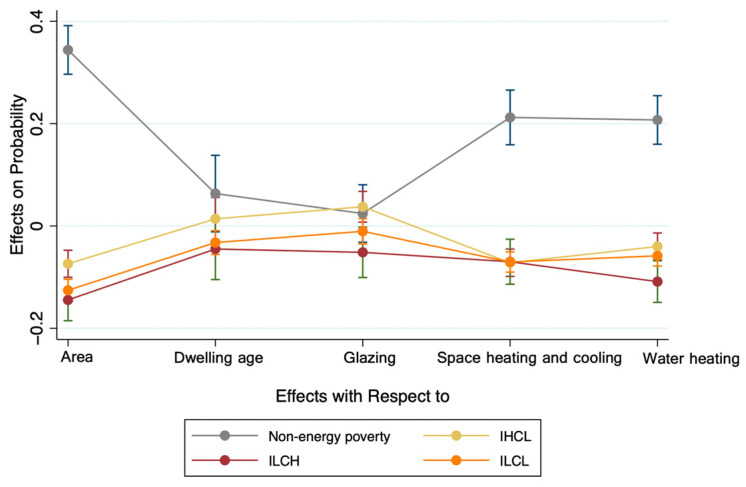
Marginal effects.

**Table 1 ijerph-19-08254-t001:** Data description.

Variables	Measure and Description
**Housing conditions**	
**Area of residence**	
Area	0 = housing in rural areas, 1 = housing in urban areas
**Basic dwelling attributes**	
Housing tenure	0 = owner-occupier, 1 = house-renter
Housing size	Living space per capita
Dwelling age	0 = dwellings built before 1980, 1 = dwelling built after 1980
**Housing insulation**	
Glazing	0 = double or triple glazing, 1 = other types
**Sunlight**	
Summer sunlight	0 = less than 8 h of sunshine in summer, 1 = more than 8 h of sunshine in summer
Winter sunlight	0 = less than 5 h of sunshine in winter, 1 = more than 5 h of sunshine in winter
**Basic energy equipment**	
Space heating and cooling	0 = no heating or air conditioning, 1 = at least one of them
Water heating	0 = no water heater, 1 = water heater
**Residents’ psychology and behavior**	
Psychological perception	The level of happiness of householder
Modern energy service preference	Sum of the number of refrigerators, freezers, washing machines and dryers
**Energy poverty**	
EPI1	0 = non-energy poverty, 1 = energy poverty
EPI2	0 = non-energy poverty, 1 = energy poverty
EPI3	0 = non-energy poverty, 1 = IHCL, 2 = ILCH, 3 = ILCL

**Table 2 ijerph-19-08254-t002:** Regression results of logit models.

	Energy Poverty Indicators				
Baseline Category: Non-Energy Poverty	EPI1	EPI2	EPI3		
	(1)	(2)	(3)	(4)	(5)
Area	0.304 ***	0.218 ***	0.246 ***	0.330 ***	0.070 ***
	(0.032)	(0.030)	(0.043)	(0.040)	(0.016)
Housing size	1.000	0.997 **	0.995 **	0.999	0.998
	(0.001)	(0.001)	(0.002)	(0.001)	(0.002)
Housing tenure	0.866	0.852	0.558 *	0.739	1.169
	(0.161)	(0.211)	(0.191)	(0.154)	(0.409)
Dwelling age	0.672 ***	0.845	1.060	0.739 *	0.553 ***
	(0.099)	(0.139)	(0.276)	(0.128)	(0.119)
Glazing	0.748 **	1.215	1.384 *	0.786 *	0.820
	(0.092)	(0.178)	(0.256)	(0.108)	(0.179)
Winter sunlight	0.844	0.783 *	0.689 **	0.794 *	0.724 *
	(0.100)	(0.108)	(0.131)	(0.109)	(0.136)
Summer sunlight	1.223 *	0.896	1.003	1.309 *	1.029
	(0.149)	(0.127)	(0.199)	(0.182)	(0.196)
Space heating and cooling	0.547 ***	0.349 ***	0.298 ***	0.523 ***	0.225 ***
	(0.058)	(0.042)	(0.051)	(0.067)	(0.037)
Water heating	0.471 ***	0.504 ***	0.461 ***	0.454 ***	0.286 ***
	(0.048)	(0.062)	(0.075)	(0.052)	(0.050)
Constant	3.324 ***	1.997 ***	1.771 *	3.009 ***	5.841 ***
	(0.585)	(0.385)	(0.544)	(0.649)	(1.503)
Observations	2402	2402	2402		
Pseudo R^2^	0.150	0.178	0.149		

Notes: Binary logit regression models are for EPI1 and EPI2, and the multinomial logit regression model is for EPI3. Under EPI3, models 3, 4 and 5 correspond to *y_i_* = 1, *y_i_* = 2 and *y_i_* = 3 respectively. The coefficients are exponentiated to obtain the relative risk ratios (RRRs). Standard errors are in parentheses. *** *p* < 0.01, ** *p* < 0.05, * *p* < 0.1.

**Table 3 ijerph-19-08254-t003:** Marginal effect.

Variables	dydx			
	Non-Energy-Poor	IHCL	ILCH	ILCL
Area	0.344 ***	−0.074 ***	−0.145 ***	−0.126 ***
Dwelling age	0.063 *	0.014	−0.045	−0.032 ***
Glazing	0.024	0.038 **	−0.051 **	−0.010
Space heating and cooling	0.212 ***	−0.072 ***	−0.070 ***	−0.070 ***
Water heating	0.207 ***	−0.040 ***	−0.109 ***	−0.058 ***

Notes: The marginal effects of housing conditions are estimated at the means of all variables. *** *p* < 0.01, ** *p* < 0.05, * *p* < 0.1.

**Table 4 ijerph-19-08254-t004:** Mediation effect.

Path A		Psychological Perception		Modern Energy Service Preference	
		Coef.	Std. Err.	Coef.	Std. Err.
	Area	−0.080 **	(0.039)	0.070 **	(0.034)
	Housing size	0.000	(0.000)	0.000	(0.000)
	Housing tenure	−0.104 *	(0.061)	−0.260 ***	(0.054)
	Dwelling age	0.048	(0.053)	0.216 ***	(0.047)
	Glazing	0.161 ***	(0.040)	0.084 **	(0.036)
	Winter sunlight	0.020	(0.041)	−0.007	(0.036)
	Summer sunlight	0.013	(0.043)	0.091 **	(0.038)
	Space heating and cooling	0.188 ***	(0.040)	0.404 ***	(0.035)
	Water heating	0.140 ***	(0.036)	0.423 ***	(0.033)
**Path B**	IHCL	Coef.	Std. Err.		
	Psychological perception	0.037	(0.099)		
	Modern energy service preference	−0.586 ***	(0.111)		
	ILCH				
	Psychological perception	−0.246 ***	(0.067)		
	Modern energy service preference	−0.355 ***	(0.081)		
	ILCL				
	Psychological perception	−0.265 ***	(0.092)		
	Modern energy service preference	−1.039 ***	(0.107)		
**Mediation test**		Z-mediation value			
		Psychological perception			
		IHCL	ILCH	ILCL	
	Area	−0.331	1.754 *	1.618	
	Housing size	0.056	−0.156	−0.153	
	Housing tenure	−0.316	1.505	1.410	
	Dwelling age	0.242	−0.854	−0.824	
	Glazing	0.360	−2.650 ***	−2.286 **	
	Winter sunlight	0.153	−0.461	−0.449	
	Summer sunlight	0.101	−0.289	−0.283	
	Space heating and cooling	0.363	−2.857 ***	−2.421 **	
	Water heating	0.359	−2.604 ***	−2.255 **	
		Modern energy service preference			
		IHCL	ILCH	ILCL	
	Area	−1.863 *	−1.799 *	−1.973 **	
	Housing size	−0.057	−0.056	−0.057	
	Housing tenure	3.525 ***	3.196 ***	4.294 ***	
	Dwelling age	−3.429 ***	−3.123 ***	−4.128 ***	
	Glazing	−2.113 **	−2.026 **	−2.269 **	
	Winter sunlight	0.200	0.199	0.203	
	Summer sunlight	−2.153 **	−2.062 **	−2.318 **	
	Space heating and cooling	−4.788 ***	−4.069 ***	−7.401 ***	
	Water heating	−4.888 ***	−4.131 ***	−7.778 ***	

Note: Path A denotes the transmission path from the housing conditions variables to the mediating variables and Path B denotes the transmission path from the mediating variables to the energy poverty categories. *** *p* < 0.01, ** *p* < 0.05, * *p* < 0.1.

## Data Availability

The data and result replication procedures used in this study can be found in the submitted replication file.

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
