# Peer review of "Linking Housing Conditions and Energy Poverty: From a Perspective of Household Energy Self-Restriction"

_ijerph, 2022, doi:10.3390/ijerph19148254_

Round 1

Reviewer 1 Report

The article addresses the relation between housing energy features and household energy poverty in China, which appears as a new and intresting perspective to investigate energy poverty in the Country.

The paper is clear and well structured. However, I would recommend accepting it after some minor revisions which would improve the overall quality:

1.      In the introduction and literature review section please try to make logical steps between the concepts more evident, as in some points it seems like sentences are not fully sequential but just a list of references.

2.      Please try to clearly divide methodology description and results, as in the result section the authors still describe some methodological part for result interpretation. E.g., the validation stage could fit more in the former part by describing it in general terms, and then described in the result section for what concerns its implementation.

3.      Figure 1: it is not clear if it refers to the Chinese situation or global situation. If the first, maybe the positioning of the figure is not fully appropriate because data collection is introduced in the following paragraph, so the figure illustrates something that has not yet been explained at that point of the text.

4.      At line 221 the authors mention the initial sample without sizing it: how big was the sample before selecting 2402 eligible observations? And what criteria were used for the selection?

5.      Line 230: why is 1980 significant to this purpose? Please explain. In Europe, for example, it indicates when energy efficiency codes entered in force.

Author Response

Dear Reviewer,

Thank you for your time spent in reading and reviewing our paper and providing these comments for us. We have made every effort to address these issues. Please see the new version of the manuscript and read our point-by-point responses for details in the “Response document”.

Thank you for your work.

Best wishes,

Yours sincerely,

Chao Feng, Ph.D.

Reviewer 2 Report

In general, the manuscript is clearly and logically written. Except English improvement in some parts of the manuscript I have no general remarks.

The detailed comments are given below:

line 45-46: "envelop enclosure [16-17] (...) of a home significantly affects the necessary energy needs of a household." - confusing style? envelope affects - in which way?

line 55-56: "households are facing affordable housing and affordable energy problems." - confusing style. I can't deduce what Authors wanted to say.

line 58-59: "suffer from poor building energy effciency" - maybe "suffer from inefficient buildings" would be better?

line 60: "to maintain the warmth of their homes." - "to maintain comfortable thermal conditions in their homes." sounds better

line 64: " household members" - occupants

line 94: "Hills (2011) identifies" - Reference number? "Hills [6] identifies"

line 101: "with house warmth," - rather with indoor thermal conditions - it is more general term, but "warmth" correlates rather with the temperature only, but this second term contains also air humidity, solar irradiance, air velocity (draughts), etc.

line 189: "equivalent income poverty threshold is 60%" - from where the value of 60% comes?

line 194: "The former are income-poor households" - maybe: "The former are low-income (poor) households" ?

Figure 1 - physical/monetary units (income and cost) 

line 215: "CGSS2015" - reference number?

Table 1: Area 0=housing in rural areas, 1=housing in urban areas

I wonder why you chose this category. Do the kind of area has a direct impact on energy poverty? Short comment will be useful for the reader.

lines 289-290: "In addition, the longer the hours of sunlight in summer, the greater the probability of energy poverty" - Intersting. Why? Maybe some technical details of these households should be provided? i suppose that this outcome is related to highly glazed (area) rural buildings with single glass windows (hence, poorly isolated). You commented this in lines 293-296, but some comments connecting these technical details could be useful.

Figure 2: A line cannot be drawn between points of different categories (Area - Dwelling age - Glazing - ...). It has no physical sense. 

"Results and discussion" section is intersting, provides valuable comments supported by the obtained results. 

Please check the reference sytle following "Instructions for Authors": https://www.mdpi.com/journal/ijerph/instructions

Author Response

(The authors gave the same response as above.)

Reviewer 3 Report

The article is well written and organized. The authors attempt to establish the link between household housing conditions and energy poverty in China from the perspective of energy self-restriction using logit and mediation models based on microdata.

Since the article tends to confirm the correlation between poverty and access to energy, I would suggest in the conclusions to briefly explain what could be possible developments of the research on the basis of more homogeneous data regarding (e.g. the average income of the sample surveyed or on the basis of surveys on a more restricted geographical area). 

At line 183 there is an error: "square root" instead of "square foot")

Author Response

(The authors gave the same response as above.)

Reviewer 4 Report

The paper is a very interesting and necessary interdisciplinary study, bringing together knowledge on housing conditions, energy poverty, psychology and behavior of householders with the aim of providing new insights on energy poverty. 

I have however some suggestions for raising its quality

General issues:

The authors use the word household to define the house but also the people living in house. Please make a clear distinction between household and householder. For example, is explored the level of happiness of the hosehold or the householder (lines 241-242)? Households (or householders?) living in rural areas are more likely to explore energy poverty – lines 286-287). Please pay attention to this aspect all over the paper content

Specific issues:

1. Introduction:

- include a reference for the statement in lines 43-45: ’ Decent, warm, comfortable homes require basic energy services, including space heating and cooling, lighting, water heating, cooking and electricity’;

- include a resumative Figure to explain how the paper is exploring the main linkages between the three fields;

- detail the working hypothesis

2. Materials and methods

- please explain the acronyms IHCL, ILCH, ILCL first time they are used - lines 206-208;

- explain why you aim to distinguish between dwellings built before 1980 and those built after 1980 – lines 230-231

- in the Methodology section, explain the use of mediation model (maybe ‚borrow’ some lines from the section 4.3 Mediation effect.

3. Results and discussion

- discuss how the results can be interpreted in perspective of previous studies and working hypothesis. The findings and their implications should be discussed in the broadest context possible and limitations of the work highlighted. Future research directions may also be mentioned.

Author Response

(The authors gave the same response as above.)
